# The Self-Identification Program (SIP): A Clinically Implemented Third-Wave CBT Deepening Dysfunctional Self-Identification in Mood Disorders

**DOI:** 10.3390/medicina61112071

**Published:** 2025-11-20

**Authors:** Martin Leurent, Déborah Ducasse

**Affiliations:** 1Therapy Center for Mood and Emotional Disorders, Department of Adult Psychiatry, La Colombière, Centre Hospitalier Universitaire, 34090 Montpellier, France; 2Institute of Functional Genomics, National Center for Scientific Research, National Institute of Health and Medical Research (INSERM), University of Montpellier, 34094 Montpellier, France; 3Department of Emergency Psychiatry and Post Acute Care, Centre Hospitalier Regional Universitaire, 34090 Montpellier, France

**Keywords:** self-concept, self-identification, mood disorders, third-wave CBT, meditation, psychotherapy process, neurophenomenology, Acceptance and Commitment Therapy, mindfulness, loving/kindness, compassion, emptiness

## Abstract

Third-wave cognitive-behavioral therapies (CBT3) have progressively shifted the focus of psychotherapy from symptom reduction to process-based and transdiagnostic mechanisms of change, emphasizing self-identification as a core dimension. Within this evolution, the Self-Identification Program (SIP) represents a conceptual and clinical advancement particularly relevant to mood disorders, where maladaptive self-identification, rumination, and self-judgment play central roles. SIP directly targets dysfunctional self-identification—the reification of transient and maladaptive mental contents as defining features of a self—through a framework integrating the three levels of CBT3: mindfulness (CBT3.1), loving/kindness and compassion (CBT3.2), and deconstructive insight into the nature of a self (CBT3.3). Theoretically, SIP aligns with dimensional psychiatry (AMPD, HiTOP, RDoC) and recent advances in behavioral linguistics (Relational Frame Theory) and psychotherapy (Process-Based Behavioral Therapy). By integrating linguistic, affective, and neuroscientific perspectives, SIP bridges contextual behavioral science and contemplative practice, offering a unified, process-based model of identity transformation. Clinically, SIP extends CBT3 beyond mindfulness and loving/kindness and/or compassion training to specifically address the mechanism by which self-identification becomes a source of suffering—namely, the mistaken identification with an independent and permanent self. In doing so, SIP provides a novel, mechanistically grounded pathway toward enduring change in depressive and bipolar spectrum disorders.

## 1. Introduction: From the Three Waves of Cbt to the Three Levels of Cbt3

CBT has earned its status as the gold standard of psychotherapy due to its commitment to empirical validation and its corrective stance toward the unscientific excesses of earlier clinical traditions [1]. Cognitive Behavioral Therapy (CBT) has evolved through three major and cumulative waves, each expanding its theoretical reach and clinical depth while retaining empirical rigor [2]. First-wave CBT (CBT1), grounded in behavioral learning theory, focused on modifying observable behaviors through conditioning and exposure [3,4,5]. Second-wave CBT (CBT2) introduced cognitive processes as mediators between situations and emotions, emphasizing the restructuring of maladaptive thoughts [6,7,8].

In the 1990s, a third wave (CBT3) emerged, shifting from content to process: rather than changing what people think, it sought to transform how they relate to inner experiences. Therapies such as Acceptance and Commitment Therapy (ACT; Hayes et al., 1999) [9], Dialectical Behavior Therapy (DBT; Linehan, 1993) [10], Mindfulness-Based Stress Reduction (MBSR; Kabat-Zinn, 1982) [11], and Mindfulness-Based Cognitive Therapy (MBCT; Segal et al., 2002) [12] emphasize mindfulness and acceptance as complementary processes rather than mere techniques [13]. In this context, mindfulness meditation provides the experiential foundation for acceptance, by fostering *self-identification as the context in which experiences arise*, rather than as their content. This shift enables individuals to recognize the inalterable and secure quality of awareness itself—a perspective that allows painful inner events to be welcomed without avoidance. In other words, acceptance emerges *not from passive tolerance*, but from a reorganization of self-experience: identifying “I” with the stable, conscious field rather than with transient mental states or evaluative narratives [2,13,14,15,16]. Through this transformation, third-wave CBTs target the *self-concept* as their central focus. Whereas first- and second-wave CBTs addressed, respectively, behaviors and cognitive content, third-wave approaches intervene upstream—at the level of identity processes: *who* experiences inner events, not only *what* is experienced [17].

Following these developments, recent theoretical work suggests that CBT3 itself can be further divided into three progressive levels [18,19]. CBT3.1 fosters self-identification with the *observing self* [20] through mindfulness meditation [21], cultivating familiarity with awareness as a safe and inalterable context for experience—the foundation of what ACT terms *Self-as-Context*. This process—long recognized as one of ACT’s six core mechanisms—has always been central to the model’s theoretical architecture [16], although its foundational role has at times been underemphasized in empirical research and clinical training. Recent advances in Relational Frame Theory reaffirm rather than redefine this function: *Self-as-Context* provides the organizing dimension that gives experiential coherence to all other ACT processes [22,23]. By shifting self-identification from the content of experience to the field in which it occurs, it reinstates the experiential depth originally intended in ACT—fostering a profound acceptance and a valued, self-congruent engagement with inner life, thereby enabling both psychological flexibility and interpersonal connectedness [24].

CBT3.2 deepens this by cultivating self-identification with the Observing self and its *natural loving kindness* [25] *and*/*or compassionate wishes* [26,27,28]. CBT3.3 involves examining and loosening identification with the self as it is usually perceived, through cultivating insight into the absence of any inherently existing self (i.e., the absence of an independent and permanent self, also called the “emptiness” of a self) [18,19,29,30,31].

Robust meta-analytic evidence supports the efficacy of CBT3.1—particularly ACT, MBSR, and MBCT—in reducing depressive and anxiety symptoms and preventing relapse in mood disorders [32,33,34,35,36,37,38,39,40,41], as well as in decreasing suicidal and self-injurious behaviors [42]. Consistently, Radically Open Dialectical Behavior Therapy (RO-DBT)—a third-wave intervention targeting maladaptive overcontrol—has also demonstrated efficacy in treatment-refractory depression, with reductions in depressive symptoms mediated by improvements in psychological flexibility and interpersonal functioning, the same acceptance-based mechanisms underlying ACT [43].

Building on this evidence, the growing interest in CBT3.1 for mood disorders naturally extends to the exploration of advanced levels—CBT3.2 and CBT3.3—that target more refined dimensions of self-identification. The Self-Identification Program (SIP), developed at the University of Montpellier in 2020, represents the first structured and manualized protocol integrating CBT3.1, CBT3.2, and CBT3.3 principles into a unified therapeutic framework. This opinion paper presents the conceptual foundations and core mechanisms of the SIP, examines its potential relevance for advancing psychotherapy in mood disorders, and summarizes its clinical implementation and preliminary evidence. It further discusses the contemporary significance of this integrative framework and outlines key implications for both clinical practice and future research.

## 2. The Structure and Core Mechanisms of the Self-Identification Program (Sip)

Like other CBT3 approaches, the Self-Identification Program (SIP) is grounded in Buddhist phenomenology, particularly the Madhyamika-Prāsaṅgika school, and draws upon traditional systems of mind training such as Lojong (training the mind; e.g., Elliott et al., 2014 [44]; Kozasa et al., 2015 [45]; MacLean et al., 2010 [46]), Lamrim (stages of the path; e.g., Bach & Guse, 2015 [26]), and Mahāmudrā (mind realization; Guillaume et al., 2020, 2024 [47,48]; Zanesco et al., 2018 [49]). Particular emphasis is placed on teachings concerning the lack of inherent existence and the causal interdependence of phenomena [50,51,52].

Note that Buddhist phenomenology typically uses the term “mind,” whereas neuroscientific literature more often refers to “consciousness” or “awareness.” In SIP, these terms are used synonymously to denote the immediate, experiential faculty of knowing—an ever-present background of awareness underlying mental events.

The SIP provides practical methods for examining and loosening identification with the self as it is usually perceived, while introducing a systematic process for cultivating self-identification with an interdependent and impermanent continuum of consciousness. This continuum is conceived as inseparable from three natural qualities of mind—Calm (an inner experience of balance, peace, and contentment), Clear (an inner experience of confidence, reassurance, and serenity, resonant with the qualities cultivated in CBT3.1), and Warm (an inner experience of worthiness, lovability, and kindness, corresponding to the qualities cultivated in CBT3.2)—as well as from the understanding of the causal relationship between mental actions and their effects. These three qualities are described as natural and spontaneous experiences of a mind (syn. consciousness) that become accessible once mental agitation subsides. Participants are not trained to “generate” these qualities but rather to recognize them as available aspects of awareness, much like the blue sky that remains present behind passing clouds. This experiential reframing supports psychological flexibility, emotional regulation, and prosocial motivation, which are hypothesized mediators of SIP’s therapeutic effects.

The SIP comprises two programs: SIP-1 and SIP-2 —whose clinical session structure is detailed in Appendix A.

SIP-1 (8 sessions of 2 h) provides a structured preparation for advanced work. It strengthens attention, interoception, and stance-taking—prerequisites for the deeper processes cultivated in SIP-2. Training is organized around eight *powers of consciousness* (pacifying, reassuring, beneficial, discerning, prioritizing, acting, creative, liberating). Each session combines brief psychoeducation with guided meditation offering an initial experience of correct self-identification. Common obstacles (e.g., “I am too damaged”) are reframed through daily practices that integrate intention and discernment in everyday life.SIP-2 (21 sessions of 2 h) delivers the core therapeutic curriculum. The first three sessions stabilize the three foundational qualities—*Calm*, *Clear*, and *Warm*—as bases for correct self-identification. The remaining sessions each address a specific self-concept arising from a perceived deficit in one or more of these qualities (e.g., agitated vs. *Calm*; vulnerable vs. *Clear*; shameful vs. *Warm*). For each, the therapist guides (i) recognition of the mistaken self-identification, and (ii) meditation on the corresponding corrective self-identification. This structured progression consolidates the basic capacities developed in SIP-1 into generalizable identity shifts that patients can embody across contexts.

Unlike CBT3.1 and CBT3.2, the SIP does not aim to “repair” an inherently existing self; rather, therapeutic transformation occurs by shifting familiarity from mistaken to correct forms of self-identification. SIP clarifies that even salutary self-identifications (e.g., “observer,” “compassionate person”) can remain subtly mistaken if reified. They become non-mistaken only when co-apprehended with the understanding of their lack of inherent existence and the understanding of the causal relationship between mental actions and their effects. This distinction highlights why SIP represents a CBT3.3: it retains the mindfulness and compassion practices of CBT3.1 and CBT3.2, yet adds the explicit deconstruction and correction of self-identification as the central therapeutic target—reinforcing the self as the key dimension of change while providing concrete methods for its transformation.

## 3. Relevance of Sip for Advancing Psychotherapy in Mood Disorders

### 3.1. Why CBT3.2 May Be Beneficial in Mood Disorders

While strong and consistent evidence supports the efficacy of CBT3.1 modalities (see Section 1 for references), CBT3.2 introduce an additional dimension that may enhance outcomes in mood disorders. By cultivating warm, caring, loving and/or compassionate self-identifications, CBT3.2 directly targets self-criticism, self-stigma, shame, and perceived social support or loneliness—mechanisms strongly implicated in both unipolar and bipolar depression (Favre & Richard-Lepouriel, 2023 [53]; Zaccari et al., 2024 [54]).

Recent meta-analyses support this proposition. A comprehensive meta-analysis of self-compassion interventions demonstrated small-to-medium reductions in depressive symptoms, with effects maintained at follow-up and concurrent improvements in anxiety and stress [55] (56 RCTs; depression SMD = 0.44 [0.31–0.57] post; 0.38 [0.14–0.61] follow-up; anxiety 0.36 [0.16–0.55] post; stress 0.43 [0.31–0.56] post; 0.23 [0.06–0.40] follow-up). Meta-analytic data on loving/kindness interventions reveal improvements in positive affect and compassion, with concurrent reductions in negative mood and psychological distress [56] (23 RCTs; vs. passive controls—mindfulness g = 0.22 [0.03–0.40], compassion 0.36 [0.06–0.65], positive affect 0.44 [0.21–0.67], negative affect 0.44 [0.14–0.73], psychological symptoms 0.36 [0.11–0.61]; effects attenuate vs. active comparators). Likewise, a systematic review and meta-analysis of Compassion-Focused Therapy in clinical populations found significant decreases in depression relative to wait-list or control conditions [57] (15 studies; depression improvements vs. wait-list/control, g ≈ 0.24–0.25).

A systematic review indicated that emotion-regulation capacity mediates the relationship between self-compassion and depression in five studies [58]. Moreover, a meta-analysis showed that self-compassion predicts better physical health, immune functioning, and sleep quality [59] (*94 samples, N ≈ 30,129; omnibus r = 0.18 between self-compassion and physical health; strongest domain effects for sleep quality and functional immunity; multi-session self-compassion interventions predicted small–to–moderate health gains*). These associations are clinically meaningful, as converging meta-analytic evidence in diagnosed depressive disorders shows that physical health burden predicts poorer remission [59], immune dysregulation is associated with poorer antidepressant response and lower remission rates [60] (*19 studies; higher baseline CRP in antidepressant non-responders SMD* = −*0.18 [*−*0.28,* −*0.08]; higher baseline IL-8 SMD* = −*0.27 [*−*0.49,* −*0.05]*) and insomnia symptoms consistently rank among the strongest negative predictors of treatment response and remission [61].

At the neurobiological level, several studies have begun to elucidate how compassion and loving/kindness practices may relate to changes in brain function and structure. A recent multimodal neuroimaging study demonstrated that Loving/Kindness Meditation integrating cognitive and behavioral components produces significant clinical improvements in major depressive disorder, associated with fronto-striatal reward-circuit modulation and increased gray matter in prefrontal regions [62]. Likewise, a meta-analysis [63] identified, in a small pooled subsample (five experiments), three consistent activation clusters for loving/kindness/compassion practice—right anterior insula, somatosensory/inferior parietal, and parieto-occipital. Complementing these findings, [64] demonstrated that loving/kindness meditation modulates hemispheric coupling patterns between attentional and emotional networks, reflecting the cultivation of balanced self-other awareness. Taken together, these findings provisionally support the mechanistic plausibility of CBT3.2 interventions. By strengthening affiliative motivation and prosocial affect, they may contribute to normalizing large-scale network dynamics (e.g., DMN and reward circuits) that are often altered in mood disorders. Within this evolving field, CBT3.2 can therefore be viewed as a process-based extension of CBT3.1, emphasizing not only acceptance and observation of experience but also the active cultivation of kindness and connectedness. Building on this emerging literature but still preliminary literature, the Self-Identification Program (SIP)—a structured framework integrating CBT3.1, CBT3.2, and CBT3.3 principles—may represent a promising next step for refining psychotherapeutic approaches to mood disorders.

### 3.2. Why Sip—Integrating Cbt3.3, Cbt3.2, and Cbt3.1—May Add Unique Value

SIP is—to our knowledge—the first CBT3.3-structured program delivered at scale and sufficiently manualized to be transferable and reproducible. It integrates approaches from CBT3.1 and CBT3.2 and goes further by providing conceptual and meditative tools to deconstruct the self as it is usually experienced. Although preliminary findings suggest that CBT3.2 may provide incremental benefits over CBT3.1 alone by enriching mindfulness-based skills with loving/kindness and/or compassionate intentions, several theoretical and mechanistic considerations indicate that SIP—integrating all three levels—could potentially support deeper and more sustainable improvements in mood disorders. We therefore propose three testable hypotheses to examine this possibility.

**Hypothesis** **1.**
*Deconstructing the “Permanent Self” May Potentiate Change Towards More Functional Self-Identifications.*


The SIP explicitly trains insight into the impermanent characteristic of a self (see Section 2), thereby dismantling the belief in a fixed personal identity. This cognitive deconstruction may facilitate clinical change by reinforcing the belief that transformation of one’s self-identification is possible. As such, SIP could amplify the efficacy of CBT3.1 and CBT3.2 mechanisms: cultivating a stable yet flexible *observing self* (CBT3.1) and fostering warm, affiliative self-relating (CBT3.2). This hypothesis aligns conceptually with evidence from the ReSource Project [65]—a large-scale, nine-month longitudinal study of secular mental training that compared three sequential modules: (a) present-moment attention and interoception, (b) socio-emotional processes like compassion and loving kindness and (c) meta-cognitive processes and perspective-taking on self and others. The study demonstrated that cultivating these distinct but complementary families of meditation-based practices produced additive and interrelated benefits across behavioral, neural, and physiological domains, suggesting potential synergistic effects when such practices are combined.

**Hypothesis** **2.**
*Deconstructing the “Independent Self” May Potentiate Reduction in Loneliness and Social Disconnection.*


The SIP explicitly trains insight into the interdependence characteristic of a self (see Section 2), thereby dismantling the profound feeling of isolation and of one’s experience being unshared or unshareable (*existential isolation*). This dimension has long been recognized in existential psychotherapy [66] and is increasingly supported by empirical research. Existential isolation differs from social loneliness: it reflects a felt separation from others at the level of meaning and inner experience, often expressed as “no one truly understands me.” Studies show that higher existential isolation predicts greater depressive symptom severity, poorer social functioning, and reduced recovery rates [67,68]. Conceptually, loosening self-identification with an autonomous, separate self directly undermines the cognitive core of loneliness and enhances perceived social connectedness. By embedding CBT3.2’s compassion training within CBT3.3’s insight into interdependence, SIP aims to correct both the *emotional* and *ontological* dimensions of disconnection. This focus is clinically relevant: a systematic review of 50 studies demonstrated that people with depression who perceive poorer social support experience worse outcomes in symptom severity, recovery, and social functioning [69].

**Hypothesis** **3.**
*Deconstructing the “independent and permanent self” may be associated with Default Mode Network (DMN) modulation.*


We hypothesize that SIP would improve DMN connectivity and reduce DMN activation at rest, reflecting a normalization of self-referential processing. The DMN—including the dorso-medial prefrontal cortex, posterior cingulate cortex, temporoparietal junction, and precuneus—is consistently implicated in self-focus, rumination, and autobiographical thought [70,71]. Abnormal DMN hyperconnectivity is a robust neural correlate of depression and relapse risk [72,73]. Meditation studies show that attentional and compassion-based practices—core to CBT3.1 and CBT3.2—reduce DMN activity [63,74]. According to Josipovic et al. (2011) [75], CBT3.3 meditations attenuate the anti-correlation between “intrinsic” (DMN) and “extrinsic” (task-positive) networks, dissolving the perceived boundaries between internal and external experience. This transient *ego-dissolution*—a phenomenological quieting of the habitual sense of self—shares neural correlates with psilocybin-induced DMN deactivation [76]. Thus, CBT3.3 meditative training could support preparatory or integrative phases of supervised psychedelic-assisted therapy, potentially even more effectively than CBT3.1 meditations which are have been considered for that purpose [77,78,79]. Collectively, these findings suggest that SIP should produce a greater post-treatment reduction in resting-state DMN activity and stronger functional reintegration than CBT3.1- or CBT3.2-based protocols, fostering more flexible and less self-centered brain dynamics in mood disorders.

## 4. Clinical Implementation and Preliminary Evidence for Sip

The Therapy Center for Mood and Emotional Disorders (CHU Montpellier) is specialized in CBT3, delivering ACT, DBT, and SIP to individuals with emotional disorders—particularly borderline personality disorder—and mood disorders, including unipolar and bipolar depression. In 2024, 427 individuals attended at least one SIP session (SIP1 or SIP2); in parallel programs that same year, 136 attended ACT and 493 attended DBT—figures that reflect a service-level investment in CBT3 generally and SIP in particular as an integrative, culminating psychotherapy. The clinical team expanded from 12 professionals in 2020 to 23 in 2025 (including nurse-therapists, psychiatrists, psychologists, and allied staff), enabling stable delivery, supervision, and scale-up.

Peer-reviewed clinical outputs to date are preliminary but informative. A case study described how SIP processes were applied to disturbed identity in borderline personality disorder (BPD), mapping therapeutic work to nine identity-diffusion categories and illustrating qualitative change across self-identification, affect regulation, and relational functioning at follow-up [30]. Design: single-case report with qualitative analysis of clinical vignettes; outcomes focused on functional identity shifts rather than symptom scales; authors note the limitation of retrospective pre-treatment reconstruction.

Beyond disorder-specific applications, a second paper tested a single 90-min SIP-inspired educational intervention addressing gender/sexual-orientation stigma in a mixed-group community sample (*n* = 172). Procedure: online conference (psychoeducation + contemplative framing) with optional Q&A; assessments at pre, immediate post, and 3-month follow-up; outcomes included external acceptance (OECD-based indicators) and internal acceptance (self-valuation, fear of negative judgment). Results indicated greater external acceptance of transgender individuals in the full sample—particularly among heterosexual participants—and reduced fear of judgment among LGBT participants; the Q&A added no detectable benefit [29].

To move from signal to substantiation, randomized controlled trials are underway at the sponsoring institution: (i) a trial comparing SIP-1 with ACT for moderate-to-severe depression, uni- or bipolar (single-center RCT; *n* = 86 [43/group]; 8 weekly 2-h group sessions; blinded outcome assessment; primary outcome = ≥50% reduction in depressive symptoms from pre- to post-treatment on BDI/MADRS; 3- and 6-month follow-ups), and (ii) a trial comparing SIP-based autobiographical writing with non-specific writing in BPD (single-center RCT; *n* = 140 [70/group]; 10 weekly 2-h sessions; primary outcome = change in Social Functioning Questionnaire [QFS] at post-intervention; 3- and 6-month follow-ups).

In addition, theoretical groundwork on Ontological Addiction Theory (OAT)—proposing that dysfunctional self-identification is a transdiagnostic driver of mental suffering—provides a published conceptual frame within which SIP’s corrective self-identification process can be situated [80]. The Ontological Addiction Scale (OAS)—a psychometric index of dysfunctional self-identification (a close construct being ego-centeredness)—has been validated in French (*n* = 492 adults with mood/emotional disorders), with strong internal consistency (OAS-24 α = 0.89), robust test–retest reliability, and a predominant single-factor structure together with coherent convergent/discriminant validity (associations with mindfulness, shame, perfectionism, self-esteem, gratitude) [81]. This complements earlier English-language validation [82] and positions the OAS as a candidate process/outcome measure for SIP trials in francophone and international settings.

## 5. Why Sip Matters Now: Three Converging Paradigm Shifts

From categorical diagnosis to process-based and dimensional care.

Psychiatry and clinical psychology are undergoing a paradigm shift from categorical to dimensional and process-based models of psychopathology. Across frameworks such as the DSM-5 Alternative Model for Personality Disorders (AMPD), ICD-11, HiTOP, and RDoC, disorders are increasingly conceptualized as graded dysfunctions in core psychological processes rather than discrete entities [83,84,85,86,87]. In psychotherapy, this same paradigm shift is embodied by Process-Based Therapy (PBT) [88,89], which seeks to replace protocol-driven, diagnosis-specific interventions with mechanism-guided, personalized approaches grounded in empirical process science. PBT integrates insights from contextual behavioral science, emotion regulation, interpersonal models, and evolutionary systems theory, forming the clinical counterpart of dimensional psychiatry.

Within this evolution, third-wave cognitive-behavioral therapies (CBT3) already illustrate such transdiagnostic orientation. Acceptance and Commitment Therapy (ACT), for instance, has shown broad efficacy across depressive, anxiety, and trauma-related conditions [33,38,41]. The SIP extends this lineage by addressing a higher-order process that regulates all others: *self-identification*. Because the self provides the frame through which experience is interpreted and enacted, dysfunctional self-identification exerts transdiagnostic influence across cognition, affect, and behavior.

2.Convergence with Contemporary Psychotherapeutic Innovations Targeting the Self.

Recent process-based innovations increasingly recognize the self as a central therapeutic process. Among these, Process-Based Behavior Therapy (PBBT) [90] represents a major advance within contextual behavioral science. Grounded in experimental work on human language and cognition, PBBT conceptualizes suffering as emerging from rigid, self-referential patterns—repetitive verbal formulations such as “*I am bad*”, “*I am unworthy*”, or “*I am anxious*”. Its clinical objective is to help clients observe, deconstruct, and recontextualize these verbal self-definitions, restoring behavioral flexibility and opening access to more adaptive forms of self-relating.

The SIP converges with this logic while extending it beyond language and behavior into the experiential domain. Like PBBT, SIP targets maladaptive self-identification as a transdiagnostic mechanism, yet it relies on a structured meditative method designed to transform how the self is *experienced* rather than merely how it is *spoken about*. Each session of SIP-2 includes a specific meditation practice aimed at revising a particular form of self-identification (e.g., “agitated,” “shameful,” or “unworthy”). Through the sequential cultivation of *Calm*, *Clear*, and *Warm* states of mind, SIP reshapes both the affective and phenomenological foundations of selfhood. This experiential training integrates the behavioral precision of contextual science with the empirically supported effects of meditation—whose efficacy in mood disorders has been established through numerous meta-analyses (See Section 1 for references related to CBT3.1, and Section 3 for references related to CBT3.2 and CBT3.3).

Together, recent behavioral innovations such as PBBT and integrative models like SIP reflect a shared paradigm shift—from regulating cognition or emotion to transforming the process of self-construction itself. SIP extends this evolution by introducing systematic meditation-based training as a method for directly reshaping the experiential foundations of selfhood.

3.Ethical and Epistemic Integration of Contemplative Practice.

Finally, SIP avoids the pitfall of “McMindfulness” [91,92,93], which warns against decontextualizing contemplative methods from their ethical and transformative roots. Its development within the secular university chair *Correct Self-Identification* brought together experienced CBT3 clinicians and a qualified Buddhist teacher, ensuring fidelity to contemplative epistemology while maintaining scientific rigor. Therapists are required to engage in personal contemplative practice, receive supervision, and uphold explicit ethical grounding. SIP thus exemplifies an ethically coherent translational model—bridging behavioral science and contemplative authenticity while safeguarding both clinical efficacy and philosophical depth.

## 6. Conclusions: Toward a Science of Correct Self-Identification

The SIP represents an evolution of third-wave cognitive-behavioral therapies (CBT3) by extending beyond attentional and mindfulness skills (CBT3.1) to systematically cultivate loving/kindness and/or compassionate intentions (CBT3.2) and to deconstruct how the self is usually constructed and stabilized moment to moment (CBT3.3).

By integrating insights from contextual behavioral science, contemplative traditions, and affective neuroscience, SIP operationalizes a multidimensional model of change that directly targets the process through which suffering is generated and maintained—namely, dysfunctional self-identification, or ontological addiction. SIP reframes psychotherapy from symptom management to identity transformation, providing a coherent clinical framework for addressing transdiagnostic processes such as rumination, self-criticism, and shame that underpin mood disorders across diagnostic spectra.

Developed through a secular collaboration between CBT3 clinicians and a qualified Buddhist teacher within the Montpellier University Chair “*Identification Correcte de Soi*” [94], SIP seeks to preserve the contemplative integrity often lost in so-called “McMindfulness” approaches [91,92,93], ensuring that scientific innovation remains grounded in both ethical coherence and transformative depth.

While this Comment is primarily intended for psychotherapy specialists, we underline the complementarity with non-psychotherapist clinicians: greater familiarity with evidence-informed, non-pharmacological options can help them encourage appropriate patient-led initiatives—particularly when rumination and self-criticism are prominent. Although SIP is not yet available outside CHU Montpellier, other third-wave CBT approaches are accessible depending on location and clinical profile (e.g., DBT for borderline personality disorder; ACT or mindfulness-based approaches for mood disorders; and CBT3.2 practices—compassion or loving/kindness—as transdiagnostic process interventions). We also encourage clinicians to remain open to nearby, available peer-support groups. In particular, programs structured on the 12-step model have demonstrated effectiveness across a wide range of conditions, with benefits extending beyond symptom reduction toward identity transformation (i.e., a shift from a self-concept of worthlessness to one of contribution and mutual aid). In a recent systematic review and meta-analysis ([95], *n* = 47 studies; 55 articles), participation in 12-step-model mutual-help groups was consistently associated with improved psychological well-being, social functioning, and quality of life and reduced substance-use severity, with pooled correlations around r ≈ 0.30 and a clear dose–response relationship between frequency of attendance and recovery outcomes.

We call for collaborative research linking behavioral, contemplative, and neural process models to refine and scale mechanisms of correct self-identification. Beyond the ongoing trials (Section 4), two pathways are prioritized. (1) Confirmatory clinical efficacy: a pragmatic RCT in borderline personality disorder comparing six months of DBT–ACT + SIP-2 versus continued DBT–ACT after an initial six-month DBT–ACT phase. The primary outcome would be reduction in suicidal ideation, and secondary endpoints focus on shame, loneliness, quality of life, peer-support participation, and inflammatory markers. (2) Process and mechanisms: longitudinal mediation studies testing whether psychological flexibility, emotion regulation, and prosocial motivation mediate the relationship between change in self-identification (indexed by the Ontological Addiction Scale and related measures) and symptom reduction. Where feasible, neural (e.g., resting-state DMN/connectivity) and immune (e.g., CRP, IL-6/IL-8) substudies, together with ecological momentary assessment, would strengthen causal inference and translational relevance.

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
