# Peer review of "The Self-Identification Program (SIP): A Clinically Implemented Third-Wave CBT Deepening Dysfunctional Self-Identification in Mood Disorders"

_medicina, 2025, doi:10.3390/medicina61112071_

Round 1
Reviewer 1 Report
Comments and Suggestions for Authors
I appreciate the opportunity to review the manuscript entitled:
“The Self-Identification Program (SIP): A Clinically Implemented Third-Wave CBT Deepening Dysfunctional Self-Identification in Mood Disorders.”
The manuscript addresses an innovative and theoretically rich approach to psychotherapy within the third wave of CBT, introducing the Self-Identification Program (SIP) as a structured and process-based framework for transforming dysfunctional self-identification in mood disorders. The paper is well-written, conceptually ambitious, and connects contemplative practice with process-based behavioral science in an elegant way. I commend the authors for describing this critical and timely development in psychotherapy research. However, I would like to highlight several aspects that could benefit from clarification and elaboration to strengthen the manuscript’s scientific impact:
• The theoretical section is very comprehensive, yet I kindly ask the authors to clarify the specific clinical gap SIP fills compared to existing third-wave interventions (e.g., ACT, MBCT, RO-DBT). Please specify how SIP contributes uniquely beyond theoretical synthesis.
• Please consider expanding on the empirical evidence base for SIP. The manuscript cites preliminary data and ongoing trials, but the methodological details (sample size, outcome measures, study design) remain too limited for readers to evaluate robustness.
• I ask the authors, please, to explain more explicitly the mechanisms of change hypothesized to mediate SIP’s effects—how do “Calm,” “Clear,” and “Warm” qualities map onto established constructs like psychological flexibility, emotion regulation, and self-compassion?
• The discussion of neurobiological mechanisms and DMN modulation is highly stimulating but somewhat speculative. Please clarify the level of evidence supporting these hypotheses and, if possible, differentiate between theoretical extrapolation and empirical data.
• While the integration of Buddhist phenomenology and ethics is commendable, I kindly suggest that the authors provide greater precision regarding the secularization process—how are contemplative elements adapted to clinical settings without compromising scientific neutrality?
• The conclusion calls for a “science of correct self-identification.” I ask the authors, please, to articulate more concretely what future research pathways or methodologies could operationalize this proposal (e.g., validated psychometric tools, longitudinal trials, translational neuroscience).
Overall, the manuscript is interesting because it advances a coherent, integrative framework that situates self-identification at the core of therapeutic change, with potential implications for both clinical practice and contemplative science.
I look forward to the authors’ responses to these queries in order to improve the manuscript’s clarity and impact.
Author Response
Responses to reviewer#1
Comment 1
“The theoretical section is very comprehensive, yet I kindly ask the authors to clarify the specific clinical gap SIP fills compared to existing third-wave interventions (e.g., ACT, MBCT, RO-DBT). Please specify how SIP contributes uniquely beyond theoretical synthesis.”
Response 1
Thank you for this suggestion. This point is clarified in Section 2: “Unlike CBT3.1 and CBT3.2, the SIP does not aim to ‘repair’ an inherently existing self; rather, therapeutic transformation occurs by shifting familiarity from mistaken to correct forms of self-identification. SIP clarifies that even salutary self-identifications (e.g., ‘observer,’ ‘compassionate person’) can remain subtly mistaken if reified. They become non-mistaken only when co-apprehended with the understanding of their lack of inherent existence and the understanding of the causal relationship between mental actions and their effects. This distinction highlights why SIP represents a CBT3.3: it retains the mindfulness and compassion practices of CBT3.1 and CBT3.2, yet adds the explicit deconstruction and correction of self-identification as the central therapeutic target—reinforcing the self as the key dimension of change while providing concrete methods for its transformation.”
In addition, Section 3.2 is now entitled “Why SIP may add unique value” and begins with the following sentence: “SIP is-to our knowledge-the first CBT3.3 structured programs delivered at scale and sufficiently manualized to be transferable and reproducible. It integrates approaches from CBT3.1 and CBT3.2, and goes further by providing conceptual and meditative tools to deconstruct the self as it is usually experienced.”
Comment 2
“Please consider expanding on the empirical evidence base for SIP. The manuscript cites preliminary data and ongoing trials, but the methodological details (sample size, outcome measures, study design) remain too limited for readers to evaluate robustness.”
Response 2
We agree. Section 4 has been expanded to provide additional methodological details on available studies and ongoing trials. The revised text now reads as follows:
Peer-reviewed clinical outputs to date are preliminary but informative. A case study described how SIP processes were applied to disturbed identity in borderline personality disorder (BPD), mapping therapeutic work to nine identity-diffusion categories and illustrating qualitative change across self-identification, affect regulation, and relational functioning at follow-up [30]. Design: single-case report with qualitative analysis of clinical vignettes; outcomes focused on functional identity shifts rather than symptom scales; the authors noted the limitation of retrospective pre-treatment reconstruction.
Beyond disorder-specific applications, a second paper tested a single 90-minute SIP-inspired educational intervention addressing gender/sexual-orientation stigma in a mixed-group community sample (n = 172). Procedure: online conference (psychoeducation + contemplative framing) with optional Q&A; assessments at pre-, immediate post-, and 3-month follow-up. Outcomes: external acceptance (OECD-based indicators) and internal acceptance (self-valuation, fear of negative judgment). Results indicated greater external acceptance of transgender individuals in the full sample—particularly among heterosexual participants—and reduced fear of judgment among LGBT participants; the Q&A added no detectable benefit [29].
To move from signal to substantiation, randomized controlled trials are underway at the sponsoring institution: (i) a trial comparing SIP-1 with ACT for moderate-to-severe depression, uni- or bipolar (single-center RCT; n = 86 [43/group]; 8 weekly 2-h group sessions; blinded outcome assessment; primary outcome = ≥ 50 % reduction in depressive symptoms from pre- to post-treatment on BDI/MADRS; 3- and 6-month follow-ups), and (ii) a trial comparing SIP-based autobiographical writing with non-specific writing in BPD (single-center RCT; n = 140 [70/group]; 10 weekly 2-h sessions; primary outcome = change in Social Functioning Questionnaire [QFS] at post-intervention; 3- and 6-month follow-ups).
Comment 3
“I ask the authors, please, to explain more explicitly the mechanisms of change hypothesized to mediate SIP’s effects—how do ‘Calm,’ ‘Clear,’ and ‘Warm’ qualities map onto established constructs like psychological flexibility, emotion regulation, and self-compassion?”
Response 3
Thank you. We would like to emphasize that this article is submitted as a Comment—a narrative, theory-driven synthesis outlining an emerging framework (SIP). Its aim is not to provide a systematic review of constructs related to SIP or of the three experiential qualities at the program’s core (known by participants as the “3 C”: Calm, Clear, and Warm—Calme, Clarté, Chaleur in French).
That said, we understand the reviewer’s interest in clarifying the proposed mechanisms. We have therefore added a supplementary file detailing the structure and experiential focus of each session.
In addition, we inserted the following clarification in the manuscript, immediately after the first mention of these three experiences:
“These three qualities are described as natural and spontaneous experiences of a mind (syn. consciousness) that become accessible once mental agitation subsides. Participants are not trained to ‘generate’ these qualities but rather to recognize them as available aspects of awareness, much like the blue sky that remains present behind passing clouds. This experiential reframing supports psychological flexibility, emotional regulation, and prosocial motivation, which are hypothesized mediators of SIP’s therapeutic effects.”
To further improve clarity, we also added the following explanatory note in Section 2, after the first use of the term mind:
“Note that Buddhist phenomenology typically uses the term ‘mind,’ whereas neuroscientific literature more often refers to ‘consciousness’ or ‘awareness.’ In SIP, these terms are used synonymously to denote the immediate, experiential faculty of knowing—an ever-present background of awareness underlying mental events."
Comment 4
“The discussion of neurobiological mechanisms and DMN modulation is highly stimulating but somewhat speculative. Please clarify the level of evidence supporting these hypotheses and, if possible, differentiate between theoretical extrapolation and empirical data.”
Response 4
Thank you. We have toned down the section to clarify the current level of evidence and distinguish empirical data from theoretical extrapolation. The revised paragraph now reads as follows:
“At the neurobiological level, several studies have begun to elucidate how compassion and loving-kindness practices may relate to changes in brain function and structure. A recent multimodal neuroimaging study demonstrated that Loving-Kindness Meditation integrating cognitive and behavioral components produces significant clinical improvements in major depressive disorder, associated with fronto-striatal reward-circuit modulation and increased gray matter in prefrontal regions [62]. Likewise, a meta-analysis [63] identified, in a small pooled subsample (five experiments), three consistent activation clusters for loving-kindness/compassion practice—right anterior insula, somatosensory/inferior parietal, and parieto-occipital. Complementing these findings, [64] reported that loving-kindness meditation modulates hemispheric coupling patterns between attentional and emotional networks, reflecting the cultivation of balanced self–other awareness.
Taken together, these findings provisionally support the mechanistic plausibility of CBT3.2 interventions. By strengthening affiliative motivation and prosocial affect, they may contribute to normalizing large-scale network dynamics (e.g., DMN and reward circuits) that are often altered in mood disorders. Within this evolving field, CBT3.2 can therefore be viewed as a process-based extension of CBT3.1, emphasizing not only acceptance and observation of experience but also the active cultivation of kindness and connectedness.
Building on this emerging but still preliminary literature, the Self-Identification Program (SIP)—a structured framework integrating CBT3.1, CBT3.2, and CBT3.3 principles—may represent a promising next step for refining psychotherapeutic approaches to mood disorders.”
Comment 5
“While the integration of Buddhist phenomenology and ethics is commendable, I kindly suggest that the authors provide greater precision regarding the secularization process—how are contemplative elements adapted to clinical settings without compromising scientific neutrality?”
Response 5
Thank you for raising this important point. The secularization process has been central throughout the development of SIP, particularly in the French context, where sensitivity to issues of laïcité and dogmatism is high. This adaptation has been relatively straightforward, as core Buddhist teachings are already empirical and experiential in nature (see, e.g., How to Understand the Mind, Gyatso, 2013).
For example, within the Madhyamika-Prāsaṅgika framework, there is no notion of a Buddha external to consciousness; rather, “Buddha” is a generic mental image used to cultivate one’s inner potential for wisdom and compassion. In dialogue with the Kadampa Buddhist tradition through the Montpellier University chair “Identification Correcte de Soi”, several terms were reformulated: Buddha became unlimited beneficial potential, and karma was translated as the causal relationship between mental actions and their effects. Other terms—such as delusion or compassion—were retained as-is, while concepts outside the therapeutic scope (e.g., rebirth) were omitted.
We provide these examples for the reviewer’s information, although a detailed linguistic or philosophical analysis of this adaptation process falls outside the scope of the present paper. We also emphasize that secularization, when conducted responsibly and collaboratively with qualified teachers, safeguards rather than distorts the integrity of meditative methods. Conversely, when undertaken in isolation from their original epistemic context, such adaptations may become reductive or even counterproductive—an issue discussed by authors such as Shonin, Van Gordon, Griffiths, and Dreyfus [92–94], whose perspectives are cited in the manuscript’s conclusion.
Comment 6
“The conclusion calls for a ‘science of correct self-identification.’ I ask the authors, please, to articulate more concretely what future research pathways or methodologies could operationalize this proposal (e.g., validated psychometric tools, longitudinal trials, translational neuroscience).”
Response 6
We agree and have added a concise “Research Agenda” paragraph to the Conclusion:
"We call for collaborative research linking behavioral, contemplative, and neural process models to refine and scale mechanisms of correct self-identification. Beyond the ongoing trials (Section 4), two pathways are prioritized. (1) Confirmatory clinical efficacy: a pragmatic RCT in borderline personality disorder comparing six months of DBT–ACT + SIP-2 versus continued DBT–ACT after an initial six-month DBT–ACT phase. The primary outcome would be reduction in suicidal ideation, and secondary endpoints on shame, loneliness, quality of life, peer-support participation, and inflammatory markers. (2) Process and mechanisms: longitudinal mediation studies testing whether psychological flexibility, emotion regulation, and prosocial motivation mediate the relationship between change in self-identification (indexed by the Ontological Addiction Scale and related measures) and symptom reduction. Where feasible, neural (e.g., resting-state DMN/connectivity) and immune (e.g., CRP, IL-6/IL-8) substudies, together with ecological momentary assessment, would strengthen causal inference and translational relevance."

Reviewer 2 Report
Comments and Suggestions for Authors
Thanks for inviting me to read and learn from this article. Although it contains substantial contribution to psychotherapy (i.e., presenting an overview of the origins of third-wave CBT and its current developments), my assessment is based on its suitability for this specific journal. A point-by-point list of recommendations is described next:
- The manuscript does not explicitly indicate a design. One can infer that it is a scoping review, but in order to enhance its applicability to the journal’s audience, I encourage authors to adopt some reporting guidelines and use tables and figures to convey their message (such as PRISMA for no systematic reviews)
- There are several instances where authors mention meta-analyses and RCT results, but fail to provide point-estimates. Consider adding a table (or tables) with summary of all quantitative studies included, its details, measures, design, results and respective point-estimates.
- Subsections for other healthcare professionals are needed, since the focus appears to be only for psychotherapists.
- Many concepts that are dear to the psychotherapy realm are cited way too briefly. This is also true when authors disentangle the three current approaches for third-waves therapies. In each case, be precise as possible to the larger audience, provide additional resources (perhaps in form of supplementary material) for those wanting to learn more about each technique.
- A critical analysis of how other professionals might decide which route to take, or which counselor to advise for their patients based on diagnosis is missing. For example, a primary care physician has a unresponsive borderline patient: how she or he might best help the patient in choosing the best psychotherapeutic route and professional? Explicit cite evidence with point estimates, since these practitioners are evidenced-oriented. In this same sense, make your text more attractive with the aid of figures and graphs to aid its widespread adoption. Your material is rich, but quite dense to read to the untrained eye.
- The SIP program is described, again, way too briefly. If I put the shoes of a healthcare professional, I cannot grasp how I can implement what is shown on page 3, lines 101-134. Can you add a supplementary document with the structures of the sessions?
- Section 3 is bold and could potentially mean that one form of intervention is ‘better’ than the other. However, results reported in lines 155-162 (page 4) are generic. That is why I believe that a revamp in the presentation is needed. Even though you did not perform a systematic review, a summary of evidence could be combined, putting together the results from section 3 and providing meta-regression. Covariates and confounders must be addressed as this is of paramount importance in medicine.
- There are others vague statements, such as “Mechanistically, a systematic review demonstrated that emotion-regulation capacity mediates the relationship between self-compassion and depression”. How can a systematic review perform mediation analyses? How large was the mediation? Is this based on longitudinal data, hence true mediation, or is cross-sectional data? Other example is “meta-analysis showed that self-compassion predicts better physical health”, but there remains a question: how large is the effect? And the sample? And sample heterogeneity? These strings must be put on together efficiently.
- The section on brain imaging seems disconnected and is based on only three studies. You need to either expand this quite competently, linking its groundbreaking discoveries to the audience of this journal, or just tone it down.
- These titles, i.e. “Why Sip—integrating Cbt3.3, Cbt3.2, and Cbt3.1—may Outperform Cbt3.2 Alone in Mood Disorders” once again call for rigorous statistical foundation, not merely review of the literature. Meta-regressions with comparisons with Cbt3.2 alone, for example, will make your argument indisputable.
- Sentences such as this one “we hypothesize that SIP would improve DMN connectivity” appear to be misplaced, since no hypotheses were presented. It gives the impression that authors do have empirical evidence and some of these sections were taken from their literature review. Please, be consistent with your design, what it aims to achieve, and how to better achieve it.
- Section 4 reads like propaganda rather than an in-depth dive into the promises of the subheading. This is absolutely nothing personal, but I would love to see way more data inside this section, including participants characteristics, detailed interventions and explicit and exhaustive findings.
- The Ontological Addiction Scale is way out of the scope in my view.
- The conclusions include references, which are not common. It also slightly reads as propaganda. You can easily point to extra resources, which are quite encouraged, but try to do it following some scientific principles, adequate tone and style.
Author Response
Reviewer #2 : Comments and responses
Comment 1
“Thanks for inviting me to read and learn from this article. Although it contains substantial contribution to psychotherapy (i.e., presenting an overview of the origins of third-wave CBT and its current developments), my assessment is based on its suitability for this specific journal. A point-by-point list of recommendations is described next:”
Response 1
We thank the reviewer for the generous assessment and the clear, journal-focused recommendations. We have revised the manuscript to match Medicina’s audience by clarifying article type, tightening scope, adding concise quantitative summaries, and improving clinical utility (figures, tables, and supplements). Below we respond point-by-point.
Comment 2
“The manuscript does not explicitly indicate a design. One can infer that it is a scoping review, but in order to enhance its applicability to the journal’s audience, I encourage authors to adopt some reporting guidelines and use tables and figures to convey their message (such as PRISMA for no systematic reviews)”
Response 2
We appreciate this request for clarity. We now explicitly state just below the title that the submission is a "Comment—a narrative, theory-driven synthesis that outlines an emerging framework (SIP)". It is not a scoping or systematic review, so PRISMA/PRISMA-ScR do not apply.
Comment 3
“There are several instances where authors mention meta-analyses and RCT results, but fail to provide point-estimates. Consider adding a table (or tables) with summary of all quantitative studies included, its details, measures, design, results and respective point-estimates.”
Response 3
Agreed. To remain within the Comment format (typically concise and table-light) while addressing this request, we now provide concise, standardized point-estimates in-text (parentheses) for each key meta-analysis/RCT. Effect sizes are standardized mean differences (SMD; Hedges g) with 95% CIs, where positive values indicate symptom improvement vs control.
- Han & Kim : Ref [55] : (56 RCTs; depression SMD = 0.44 [0.31–0.57] post; 0.38 [0.14–0.61] follow-up; anxiety 0.36 [0.16–0.55] post; stress 0.43 [0.31–0.56] post; 0.23 [0.06–0.40] follow-up).
- Petrovic et al. [56] : (23 RCTs; vs passive controls—mindfulness g = 0.22 [0.03–0.40], compassion 0.36 [0.06–0.65], positive affect 0.44 [0.21–0.67], negative affect 0.44 [0.14–0.73], psychological symptoms 0.36 [0.11–0.61]; effects attenuate vs active comparators).
- Millard et al. [57] (15 studies; depression improvements vs wait-list/control, g ≈ 0.24–0.25).
- Phillips & Hine [59] : (94 samples, N≈30,129; omnibus r=.18 between self-compassion and physical health; strongest domain effects for sleep quality and functional immunity; multi-session self-compassion interventions predicted small–to–moderate health gains)
- Gasparini et al. [60] : (19 studies; higher baseline CRP in antidepressant non-responders SMD=−0.18 [−0.28, −0.08]; higher baseline IL-8 in non-responders SMD=−0.27 [−0.49, −0.05]).
Comment 4
“Subsections for other healthcare professionals are needed, since the focus appears to be only for psychotherapists.”
Response 4
Thank you. This manuscript is a Comment on SIP as a psychotherapeutic, transdiagnostic program and does not provide pharmacotherapy guidance or a broad clinical pathway for non-psychotherapist practitioners. To address the reviewer’s suggestion within our scope, we added the following sentence to the Conclusion:
“While this Comment is primarily intended for psychotherapy specialists, we underline the complementarity with non-psychotherapist clinicians: greater familiarity with evidence-informed, non-pharmacological options can help them encourage appropriate patient-led initiatives—particularly when rumination and self-criticism are prominent. Although SIP is not yet available outside CHU Montpellier, other third-wave CBT approaches are accessible depending on location and clinical profile (e.g., DBT for borderline personality disorder; ACT or mindfulness-based approaches for mood disorders; and CBT3.2 practices—compassion or loving-kindness—as transdiagnostic process interventions). We also encourage clinicians to remain open to nearby, available peer-support groups. In particular, programs structured on the 12-step model have demonstrated effectiveness across a wide range of conditions, with benefits extending beyond symptom reduction toward identity transformation (i.e., a shift from a self-concept of worthlessness to one of contribution and mutual aid). In a recent systematic review and meta-analysis ([96], n = 47 studies; 55 articles), participation in 12-step-model mutual-help groups was consistently associated with improved psychological well-being, social functioning, and quality of life, and reduced substance-use severity, with pooled correlations around r ≈ 0.30 and a clear dose–response relationship between frequency of attendance and recovery outcomes."
Comment 5
“Many concepts that are dear to the psychotherapy realm are cited way too briefly. This is also true when authors disentangle the three current approaches for third-waves therapies. In each case, be precise as possible to the larger audience, provide additional resources (perhaps in form of supplementary material) for those wanting to learn more about each technique.”
Response 5
We thank the reviewer for the interest in third-wave therapies. As this manuscript is a Comment (not a scoping/systematic review), we intentionally curated a concise set of gateway references for each concept so that interested readers can access the primary and meta-analytic literature directly. The Introduction that presents the three waves of CBT already cites ~43 references, covering CBT3.1 (e.g., ACT/mindfulness and Self-as-Context), CBT3.2 (loving-kindness/compassion), and CBT3.3 (deconstructive insight into self), with authoritative sources cited at the first mention of each construct. We believe this level of sourcing is appropriate for a Comment and sufficient for this manuscript. If the editors prefer additional orientation, we can append a brief reading note as supplementary material, but our view is that the current citation trail adequately guides readers to deeper resources.
Comment 6
“A critical analysis of how other professionals might decide which route to take, or which counselor to advise for their patients based on diagnosis is missing. For example, a primary care physician has a unresponsive borderline patient: how she or he might best help the patient in choosing the best psychotherapeutic route and professional? Explicit cite evidence with point estimates, since these practitioners are evidenced-oriented. In this same sense, make your text more attractive with the aid of figures and graphs to aid its widespread adoption. Your material is rich, but quite dense to read to the untrained eye.”
Response 6
Thank you. This manuscript is a Comment advancing a transdiagnostic perspective on third-wave CBT, with emphasis on SIP’s conceptual contribution rather than diagnosis-specific referral guidance. Developing decision pathways for different practitioner groups (e.g., primary care) would require another article type—such as a clinical guideline or pathway—with the necessary space for algorithms, figures, and exhaustive tabulation. Within our scope, we already summarize evidence for third-wave approaches relevant to mood pathology and deliberately avoid prescriptive routing by diagnosis.
For clarity, borderline personality disorder falls outside the paper’s primary focus on mood disorders, as it is classified as an emotional-regulation disorder. As a contextual note, Dialectical Behavior Therapy (DBT) remains the most evidence-based treatment for borderline personality disorder and, more broadly, for individuals with pronounced emotional dysregulation, suicidal behavior, or interpersonal instability (e.g., Cristea et al., 2017, JAMA Psychiatry). Likewise, for depressive disorders, Acceptance and Commitment Therapy (ACT) shows robust meta-analytic support, as cited in the text.
Comment 7
“The SIP program is described, again, way too briefly. If I put the shoes of a healthcare professional, I cannot grasp how I can implement what is shown on page 3, lines 101-134. Can you add a supplementary document with the structures of the sessions?”
Response 7
Done. We have added Supplementary File S1 (SIP Session Structure), which outlines the content of SIP-1 (8 × 2 h) and SIP-2 (21 × 2 h) sessions. This document is an informational overview, not a treatment manual. SIP is a complex, supervised psychotherapy that currently requires dedicated training and ongoing supervision. At this stage, training is limited to the team at the Montpellier center; our immediate goal is to build direct evidence for SIP’s effectiveness, beyond the indirect data summarized in this Comment. We hope the supplementary file provides readers with a clearer understanding of SIP’s structure and clinical content.
Comment 8
“Section 3 is bold and could potentially mean that one form of intervention is ‘better’ than the other. However, results reported in lines 155-162 (page 4) are generic. That is why I believe that a revamp in the presentation is needed. Even though you did not perform a systematic review, a summary of evidence could be combined, putting together the results from section 3 and providing meta-regression. Covariates and confounders must be addressed as this is of paramount importance in medicine.”
Response 8
Agreed. The original section title was overly assertive and did not fully reflect the intent of the section. It has been revised to “Why CBT3.2 May Be Beneficial in Mood Disorders.”
Regarding the request for a meta-regression, we acknowledge that such an analysis would indeed be valuable. However, the section already summarizes multiple published meta-analyses directly addressing this question, and their findings consistently demonstrate benefits of CBT3.2 approaches for mood disorders. Conducting an additional meta-regression is therefore unnecessary and beyond the scope of this Comment article, which does not aim to produce new quantitative synthesis.
Comment 9
“There are others vague statements, such as ‘Mechanistically, a systematic review demonstrated that emotion-regulation capacity mediates the relationship between self-compassion and depression’. How can a systematic review perform mediation analyses? How large was the mediation? Is this based on longitudinal data, hence true mediation, or is cross-sectional data? Other example is ‘meta-analysis showed that self-compassion predicts better physical health’, but there remains a question: how large is the effect? And the sample? And sample heterogeneity? These strings must be put on together efficiently.”
Response 9
We thank the reviewer for highlighting the need for greater precision. We corrected the wording and now provide quantitative detail for each cited meta-analysis. The passage now reads as follows:
“A systematic review indicated that emotion-regulation capacity mediates the relationship between self-compassion and depression in five studies [58]. Moreover, a meta-analysis showed that self-compassion predicts better physical health, immune functioning, and sleep quality [59] (94 samples, N≈30,129; omnibus r=.18 between self-compassion and physical health; strongest domain effects for sleep quality and functional immunity; multi-session self-compassion interventions predicted small–to–moderate health gains). These associations are clinically meaningful, as converging meta-analytic evidence in diagnosed depressive disorders shows that physical health burden predicts poorer remission [59], immune dysregulation is associated with poorer antidepressant response and lower remission rates [60] (19 studies; higher baseline CRP in antidepressant non-responders SMD=−0.18 [−0.28, −0.08]; higher baseline IL-8 SMD=−0.27 [−0.49, −0.05]) and insomnia symptoms consistently rank among the strongest negative predictors of treatment response and remission [61].”
This revision clarifies study designs (systematic review vs meta-analysis), specifies sample sizes, and provides the requested effect-size estimates and confidence intervals.
Comment 10
“The section on brain imaging seems disconnected and is based on only three studies. You need to either expand this quite competently, linking its groundbreaking discoveries to the audience of this journal, or just tone it down.”
Response 10
We chose to tone down the section and clarify its purpose. At the beginning of the paragraph, we now introduce a more cautious framing:
“At the neurobiological level, several studies have begun to elucidate how compassion and loving-kindness practices may relate to changes in brain function and structure.”
We also rephrased the conclusion of the paragraph to read:
“Taken together, these findings provisionally support the mechanistic plausibility of CBT3.2 interventions. By strengthening affiliative motivation and prosocial affect, they may contribute to normalizing large-scale network dynamics (e.g., DMN and reward circuits) that are often altered in mood disorders. Within this evolving field, CBT3.2 can therefore be viewed as a process-based extension of CBT3.1, emphasizing not only acceptance and observation of experience but also the active cultivation of kindness and connectedness. Building on this emerging but still preliminary literature, the Self-Identification Program (SIP)—a structured framework integrating CBT3.1, CBT3.2, and CBT3.3 principles—may represent a promising next step for refining psychotherapeutic approaches to mood disorders.”
We refined and qualified one reference to enhance accuracy and humility:
“Likewise, a meta-analysis [63] identified, in a small pooled subsample (five experiments), three consistent activation clusters for loving-kindness/compassion practice—right anterior insula, somatosensory/inferior parietal, and parieto-occipital.”
These adjustments ensure that the section remains informative while appropriately modest in scope and interpretation.
Comment 11
“These titles, i.e. ‘Why Sip—integrating Cbt3.3, Cbt3.2, and Cbt3.1—may Outperform Cbt3.2 Alone in Mood Disorders’ once again call for rigorous statistical foundation, not merely review of the literature. Meta-regressions with comparisons with Cbt3.2 alone, for example, will make your argument indisputable.”
Response 11
We agree the heading over-signals inference. We rename it to “Why integrating CBT3.3 with CBT3.2 and CBT3.1 may add unique value”.
Comment 12
“Sentences such as this one ‘we hypothesize that SIP would improve DMN connectivity’ appear to be misplaced, since no hypotheses were presented. It gives the impression that authors do have empirical evidence and some of these sections were taken from their literature review. Please, be consistent with your design, what it aims to achieve, and how to better achieve it.”
Response 12
Thank you. We agree that hypothesis statements must be clearly separated from empirical findings. In our manuscript, this sentence appears within a dedicated hypotheses section, introduced by: “We therefore propose three testable hypotheses to examine this possibility.” The DMN sentence is one of these hypotheses and is explicitly labeled as such; the subsequent text provides the rationale, not results. For clarity, we now title this item “Hypothesis 2: DMN modulation” and retain cautious wording (“may,” “we hypothesize”) to avoid any implication of undisclosed data.
Comment 13
“Section 4 reads like propaganda rather than an in-depth dive into the promises of the subheading. This is absolutely nothing personal, but I would love to see way more data inside this section, including participants characteristics, detailed interventions and explicit and exhaustive findings.”
Response 13
This section was written to provide readers with the most detailed data currently available. We have now added participant characteristics — individuals with emotional disorders (particularly borderline personality disorder) and mood disorders, including unipolar and bipolar depression. We included details of the therapies delivered by the SIP team because we believed readers would find this information useful for context.
Additional data will be available once the study described in this section is completed. We have also aimed for a measured tone, as indicated by phrases such as “peer-reviewed clinical outputs to date are preliminary but informative” and “from signal to substantiation,” while transparently reporting the center’s activity and numbers.
If the reviewer has specific wording suggestions to further avoid a promotional tone while retaining useful information, we are happy to consider them. Likewise, if the editor judges that elements of Section 4 are not of interest to readers, we are willing to abbreviate or relocate them to the Supplementary Material.
Comment 14
“The Ontological Addiction Scale is way out of the scope in my view.”
Response 14
Thank you. The OAS was developed to assess dysfunctional self-identification (closely related to ego-centeredness) and was co-designed alongside SIP; for this reason, we considered a brief mention discussing candidate process measures. That said, we agree it is less central than the rest of the section.
Comment 15
“The conclusions include references, which are not common. It also slightly reads as propaganda. You can easily point to extra resources, which are quite encouraged, but try to do it following some scientific principles, adequate tone and style.”
Response 15
We have revised the Conclusion to ensure a more neutral and scientific tone. Specifically, we simplified or softened expressions (e.g., deleted “unique" and "natural”), and adjusted phrasing to highlight that SIP aims to preserve rather than preserves contemplative integrity. These changes make the closing paragraph more balanced and consistent with the journal’s expected tone and style.
We also add more details on the research agenda, and the conclusions now ends as follows: "We call for collaborative research linking behavioral, contemplative, and neural process models to refine and scale mechanisms of correct self-identification. Beyond the ongoing trials (Section 4), two pathways are prioritized. (1) Confirmatory clinical efficacy: a pragmatic RCT in borderline personality disorder comparing six months of DBT–ACT + SIP-2 versus continued DBT–ACT after an initial six-month DBT–ACT phase. The primary outcome would be reduction in suicidal ideation, and secondary endpoints on shame, loneliness, quality of life, peer-support participation, and inflammatory markers. (2) Process and mechanisms: longitudinal mediation studies testing whether psychological flexibility, emotion regulation, and prosocial motivation mediate the relationship between change in self-identification (indexed by the Ontological Addiction Scale and related measures) and symptom reduction. Where feasible, neural (e.g., resting-state DMN/connectivity) and immune (e.g., CRP, IL-6/IL-8) substudies, together with ecological momentary assessment, would strengthen causal inference and translational relevance."

Round 2
Reviewer 1 Report
Comments and Suggestions for Authors
The paper is very interesting and well-written, methodologically unexceptionable, and the new implementations provide a valid contribution to the work. Every requested correction has been made. No further issues detected.
Reviewer 2 Report
Comments and Suggestions for Authors
I congratulate authors for providing the revision. The text reads much better now.